# Live-imaging rate-of-kill compound profiling for Chagas disease drug discovery with a new automated high-content assay

**Nina Svensen, Susan Wyllie, David W. Gray, Manu De Rycker** *

Wellcome Centre for Anti-Infectives Research, School of Life Sciences, University of Dundee, Dundee, United Kingdom

* m.derycker@dundee.ac.uk

## Abstract

Chagas disease, caused by the protozoan intracellular parasite *Trypanosoma cruzi*, is a highly neglected tropical disease, causing significant morbidity and mortality in central and south America. Current treatments are inadequate, and recent clinical trials of drugs inhibiting CYP51 have failed, exposing a lack of understanding of how to translate laboratory findings to the clinic. Following these failures many new model systems have been developed, both *in vitro* and *in vivo*, that provide improved understanding of the causes for clinical trial failures. Amongst these are *in vitro* rate-of-kill (RoK) assays that reveal how fast compounds kill intracellular parasites. Such assays have shown clear distinctions between the compounds that failed in clinical trials and the standard of care. However, the published RoK assays have some key drawbacks, including low time-resolution and inability to track the same cell population over time. Here, we present a new, live-imaging RoK assay for intracellular *T. cruzi* that overcomes these issues. We show that the assay is highly reproducible and report high time-resolution RoK data for key clinical compounds as well as new chemical entities. The data generated by this assay allow fast acting compounds to be prioritised for progression, the fate of individual parasites to be tracked, shifts of mode-of-action within series to be monitored, better PKPD modelling and selection of suitable partners for combination therapy.

## Author summary

Chagas disease is caused by the single cell protozoan parasite *Trypanosoma cruzi*. Millions of people suffer from this disease in central and south America, which frequently causes heart disease and can result in death. Chagas disease is classified as a neglected tropical disease due to the lack of investment in development of new medicines. The currently available medicines are inadequate as they require long treatments, often with severe side-effects. To develop new medicines, it is critical to build laboratory assays and tools that help predict the ability of new compounds to cure patients. Rate-of-kill assays measure how quickly compounds can kill parasites, providing a route to differentiate promising compounds from poor ones. Here, we describe development of an advanced rate-of-kill

**Data Availability Statement:** All relevant data are within the manuscript and its Supporting Information files.

**Funding:** This work was funded by Wellcome Trust (https://wellcome.org/) awards 203134/Z/16/Z (NS, DWG), 204672/Z/16/Z (MDR, DWG) and 218448/Z/19/Z (SW). The funder did not play a role in the study design, data collection and analysis, decision to publish, or preparation of the manuscript.

**Competing interests:** The authors have declared that no competing interests exist.

assay that, unlike existing assays, can monitor the same cell population over the duration of compound treatment. Using live-cell microscopy, parasite-infected host cells and their response to compound treatment can be continuously monitored. This enables better defined rate-of-kill profiles to be produced, in turn allowing better informed decisions on subsequent compound progression. Here, we report the live-imaging rate-of-kill profiles for several key compounds, including current drugs and compounds in clinical development.

## Introduction

Chagas disease presents a serious and ongoing health problem in Latin America. It affects approximately 6–7 million people worldwide, with an estimated 1.2 million suffering from Chagasic cardiopathy [1] and results in more than 10,000 deaths annually [2]. Currently, the nitroaromatic compounds benznidazole and nifurtimox are the only drugs approved for treatment of Chagas disease. Both drugs cause severe side-effects and are not consistently efficacious for the treatment of chronic Chagas disease [3], the most prevalent clinical presentation. Several new compounds, all CYP51 inhibitors, have been progressed to human clinical trials but failed to show efficacy [4,5]. As a result, no new drugs have been developed for treatment of Chagas disease for more than 30 years, and very few compounds are currently progressing through clinical development (notably fexinidazole[6]). Thus, there is an urgent need to develop new drugs with novel modes-of-action as well as improved efficacy and safety profiles.

A major challenge in drug discovery for Chagas disease is identifying ways to improve the translation from *in vitro* assays to efficacy in animal models of disease. Standard endpoint, cell-based screening assays yield very little information beyond potency and, as such, provide limited guidance for compound prioritisation, design of *in vivo* dosing regimens, understanding of pharmacokinetic/pharmacodynamic (PKPD) relationships and prediction of appropriate compound combinations.

Determining how fast a compound can kill intracellular parasites *in vitro* and the corresponding effect-versus-time profile can provide valuable additional information. Rate-of-kill profiles are dependent upon a combination of compound mode-of-action (MoA) and the compound's specific physicochemical properties. Thus, these profiles can provide information that allows prioritisation of fast acting compounds [7], better PKPD modelling [8], monitoring changes in MoA across a compound series and assessing compound combinations for improved rate of parasite clearance. However, the relatively low time-resolution in current rate-of-kill assays [9,10], with parasites counted every 24h, results in low resolution rate-of-kill curves, with poorly defined lag phases and maximum rate-of-kill. A further drawback of current assays is that they measure parasite levels in distinct samples at each timepoint, increasing the inherent variability of the assay. A more advanced and improved live-imaging assay was recently reported, however, this assay is still unable to track the same cell population for the duration of compound exposure. Instead distinct cell populations are imaged for 24 hour intervals [11], complicating the subsequent analysis of the rate-of-kill profile.

Here, we present the development of a live-imaging rate-of-kill assay for *T. cruzi* that offers high and tuneable temporal resolution. We present data for clinical compounds as well as several chemical series in development and discuss the additional value provided by this assay for Chagas disease drug discovery.

## Methods

### Reagents

Benznidazole, nifurtimox, and formaldehyde were obtained from Sigma. Posaconazole was purchased from Sequoia Research Products. All other compounds were synthesised in house as previously published [12–17].

### Parasite and mammalian cell lines

Vero cells (ECCAC 84113001) and H9C2 cells (ECCAC 88092904) were screened for *Mycoplasma* contamination and maintained in DMEM Glutamax medium (Thermo Fisher) supplemented with 10% (v/v) foetal calf serum (FCS; Hyclone; culture media) at 37˚C in presence of 5% $CO_2$. *T. cruzi* amastigotes (Silvio X10/7 A1 [18]) were propagated in Vero cells, as previously described [19]. On a weekly basis, trypomastigotes emerging from Vero cells were harvested and used to infect a new Vero monolayer with multiplicity of infection (MOI) 1.5. Epimastigotes were grown at 28˚C in RTH/FBS [RPMI 1640 medium supplemented with trypticase (0.4%), 25 µM hemin, 17 mM Hepes (pH7.4), and 10% (v/v) heat-inactivated foetal bovine serum (FBS, PAA Laboratories; now GE Healthcare)] [20].

### Generation of *T. cruzi* X10/7 E2-Crimson cell line

The E2-Crimson gene sequence (Evrogen, E2-Crimson-N) was submitted to GenScript (Piscataway, NJ) for OptimumGene codon optimization to facilitate expression in *T. cruzi spp*. The resulting synthetic gene (sequence in S1 Text) was ligated into pUC57 via XbaI sites at the 5' and 3' ends of the gene. The pUC57-eGFP construct was digested with XbaI and the resulting fragment was ligated into the XbaI-digested *T. cruzi* expression vector pTREX, resulting in the pTREX-E2Crimson plasmid. The plasmid was then linearised through digestion with NheI in preparation for transfection. Mid-log *T. cruzi Silvio* X10/7 A1 epimastigotes were transfected with 10 µg of linearised pTREX-E2Crimson using the Human T-cell Nucleofector kit and Amaxa Nucleofector electroporator (program U-033). Following transfection, the cells were left to recover for 24 h in RTH media prior to selection with neomycin (100 µg/mL). Neomycin-selected epimastigotes were allowed to reach stationary phase and differentiate to trypomastigotes. These differentiated parasites were used to infect monolayers of Vero cells and then cycled through these cells, as described above. Cloned cell lines were generated via infection of Vero cells ($10^3$ cells/well, 384-well assay plate) by limiting dilution of the *Tc*-X10/7-E2Crimson trypomastigotes (0.5 trypomastigotes/well) and infected cells were maintained in culture media supplemented with neomycin (200 µg/mL). Clonal parasites expressing the highest levels of E2Crimson were then selected using fluorescence microscopy (Incucyte Zoom, Essen BioScience) and analysed by flow cytometry as follows. Parasites were fixed using formaldehyde (1% final concentration) for 30 min and collected by centrifugation in a minispin tabletop centrifuge (10,000 rpm for 5 min). The cells were resuspended in PBS and E2Crimson expression levels were monitored using a 561nm laser and a 710/50 bandpass filter on a CytoFLEX S (Beckman Coulter).

### Generation of eGFP-labelled H9C2 cell line

A H9C2 cell line stably expressing nuclear restricted eGFP (NucLight Green (Sartorius)) was generated as follows: low passage H9C2 cells were grown to 10% confluence in a 6-well plate and infected with NucLight Green (Sartorius) by adding eGFP-lentivirus particles at an MOI of 1 (100 µL, $10^6$ TU/mL) and Polybrene (8 µg/ml) to 1 mL fresh culture media, followed by incubation for 24 h. The media was replenished and the cells were incubated for a further 48 h.

The cells were then detached with trypsin (0.5% in PBS) for 5 min at 37˚C/5% $CO_2$, plated in 6 well plates at 30% confluence in culture media (3mL) and allowed to attach for 24 h. Then, puromycin was added to the media to a final concentration of 3 µg/mL. Drug selection was continued for 5 days. Next, cells were detached with trypsin, resuspended in DMEM media supplemented with 1% FBS ($10x10^6$/ml) and the top 10% eGFP-positive cells were collected by fluorescence-activated cell sorting and immediately seeded in a 6-well plate in 3 ml of culture media supplemented with 1 µg/mL puromycin.

## Live-imaging *T. cruzi* rate-of-kill assay

For the rate-of-kill assay infection conditions were chosen so that a relatively high level of infection (> 60%) was achieved and trypomastigote egress during the assay was limited. First, *Tc*-X10/7-E2Crimson trypomastigotes that had been cycled through Vero cells at least 8 times, were used to infect H9C2-eGFP cells in 40 mL culture media in a T175 filter top tissue culture flasks (Eppendorf) using an MOI of 15 ($3 x 10^7$ parasites to $2 x 10^6$ host cells) for 24 h at 37˚C in presence of 5% $CO_2$. Residual extracellular trypomastigotes were washed away with fresh culture media (7 x 12 mL) and the infected H9C2-eGFP cells incubated for 6 h at 37˚C / 5% $CO_2$. Infected H9C2-eGFP cells were harvested by trypsinisation and resuspended in Fluoro-Brite DMEM media (Thermo Fisher) supplemented with GlutaMax (Thermo Fisher) and 10% (v/v) FCS (assay media) at $8 x 10^4$ cells/mL. Infected H9C2-eGFP cells were plated into 384-well assay plates containing the compounds to be tested at 8,000 cells per well in assay media (100 µL per well), which resulted in a near confluent monolayer. The plates were placed in the IncuCyte Zoom (Sartorius) instrument and imaged with a 20x Nikon objective, Dual Colour Model 4459 Filter module, phase channel, green channel (600 ms), and red channel (900 ms) every 3 hours for 5 days. The IncuCyte Zoom allows acquisition of one image per well in a 384-well plate, roughly in the middle of the well. The area covered by each image is 535,500 µm$^2$.

## Image analysis algorithm

Images were analysed using the Columbus high-content image analysis system (Perkin Elmer) using raw images and built-in algorithms know as building blocks. The algorithm for the standard rate-of-kill assay first identified the eGFP-positive H9C2 nuclei using the Columbus 'find nuclei' building block (each nucleus is detected as a region on the image having a higher eGFP intensity than the surrounding area), followed by demarcation of the H9C2 cytoplasm using the Columbus 'find cytoplasm' building block (identifies region with higher intensity than background surrounding the nucleus, low levels of cytoplasmic eGFP are used to segment the cytoplasm). Intracellular E2Crimson-positive *Tc*-X10/7 amastigotes were segmented using the Columbus 'find spots' building block, which detects small regions on the image having a higher E2-Crimson intensity than the surrounding area. These were confined to the host cytoplasm to exclude extracellular trypomastigotes. To reduce false positive amastigote identification, filters were applied (mean intensity >11, parasite area >11 and <90 µm$^2$ and parasite roundness > 0.65). Finally, the percentage of infected host cells was calculated by quantifying the number of infected host cells (using "Select Population" building block that identifies host cells with one or more intracellular parasites) and expressing this as a percentage of total H9C2 cells in the image. Thus, this algorithm reported the total number of host cells, total number of red parasites, mean number of red parasites per host cell and mean number of percent infected host cells.

   The maximum rate-of-kill was measured by calculating the slope of a linear regression curve with an R$^2$>0.95 fitted to at least 4 points on the most linear and steepest part of the

rate-of-kill curve. The lag phase was measured by visual inspection of the rate-of-kill curves and estimating the flat portion of the curve before a decrease in percent infected cells is observed. Compound potency was determined using data from the 120 hours timepoint (post start of treatment) as previously described[9].

## Cell replication assessments

Cell replication was assessed under assay conditions following the protocol for the standard rate-of-kill assay, described above. Image analysis was as above and total number of H9C2-eGFP cells and total number of *Tc*-X10/7-E2Crimson intracellular amastigotes in the field of view were counted and plotted against time. The doubling time was calculated by plotting the logarithm of the mean number of amastigotes per well versus time during exponential growth and fitting the resulting curve with the exponential growth formula (see below). The mean amastigote growth rate was calculated across 24 wells (n = 24 technical replicates).

$N(t) = c * 2^{t/d}$, where $N(t)$ = number of parasites at time $t$, $d$ = doubling time, and $c$ = initial number of parasites.

## Results

### Development of a live-imaging assay for *Trypanosoma cruzi*

We set out to develop a live-imaging assay of intracellular *T. cruzi* based on expression of fluorescent proteins as a convenient way to track live parasites and host cells using fluorescence microscopy. We generated a *T. cruzi* cell line expressing the far-red fluorescent protein E2-crimson and H9C2 cardiomyocytes (host cells) expressing nucleus-restricted eGFP. H9C2 were chosen as host cells due to their suitability for imaging (e.g. large flat cells). E2-crimson expression in *T. cruzi* was verified by flow cytometry (**S1 Fig**). Live imaging was performed on a microscope housed within a tissue culture incubator. An outline of the assay is shown in **Fig 1A**, and full details are provided in the methods section. To reduce background fluorescence, a culture media specifically designed for fluorescence imaging with reduced levels of fluorescent components (FluoroBrite-DMEM, ThermoFisher) was used. We first determined growth curves for the host cells, using an image analysis algorithm that identifies fluorescent host cell nuclei. **Fig 1C (left panel)** shows that host cell growth was minimal in the FluoroBrite media, and that viability was maintained for at least four days post plating in the absence of parasites. The lack of host cell growth is advantageous since cell division of infected cells can confound interpretation of infection measurements. Next, we performed imaging of infected cells. **S1 Movie** shows that parasite burden increases with time in vehicle treated cells, eventually resulting in the rupture of host cells and release of trypomastigotes (on days 3 and 4 post plating). For analysis of infection dynamics, an image analysis algorithm was developed that uses the fluorescent signals to count the total number of parasites and host cells, as well as percentage of infected host cells (**Fig 1B**). The initial level of infection of the H9C2 cells, calculated as the number of infected host cells divided by the total number of host cells expressed as percentage, was high, with 69 ± 12% (N = 20) cells infected. Parasite numbers and the percentage of infected cells increased with time, reaching a maximum approximately 50 h post plating (**Fig 1C, middle and right panels).** As expected, the initial increase in parasite numbers is followed by a decrease in both parasite and host cell numbers due to parasite egress, which becomes clearly observable approximately 60-70h post plating (~90 – 100h post infection). Based on these data a parasite doubling time of 14.7 ± 1.4 h was calculated in the exponential growth phase.

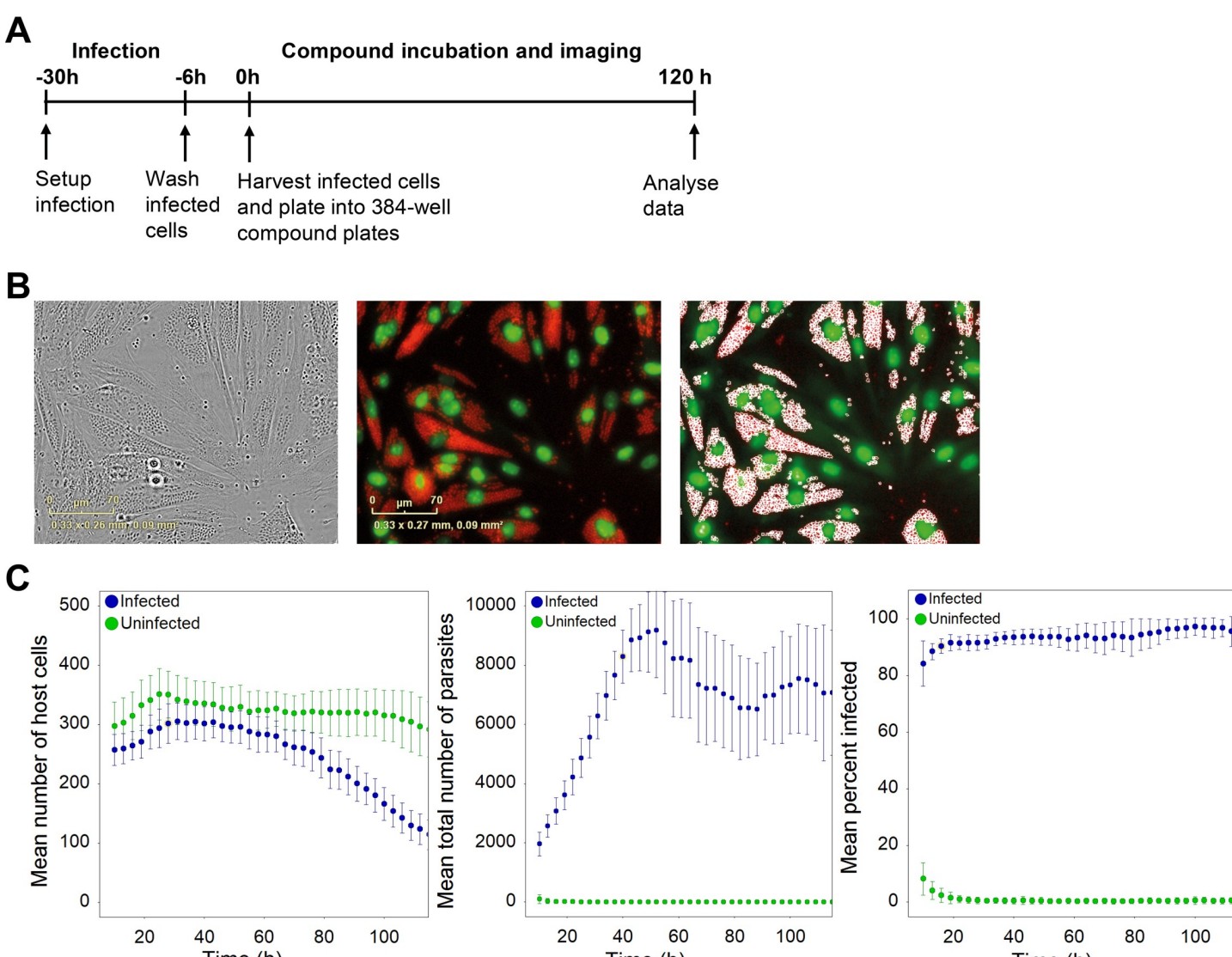

**Fig 1.** Rate-of-kill assay development: a) Rate-of-kill assay outline; b) Live cell imaging of *Tc*-X10/7-E2Crimson-infected H9C2-eGFP cells (left panel: brightfield, middle panel: fluorescence image) and image algorithm segmentation segmentation/quantification of E2Crimson-labeled parasites (right panel, segmented parasites are circled white); c) time-course data. Left panel: mean number of *Tc*-X10/7-E2Crimson-infected H9C2-eGFP cells versus time. Middle panel: total number of intracellular amastigotes detected by the image analysis algorithm per well versus time. Right panel: percent infected host cells versus time. The small number of infected cells seen at early timepoints in the uninfected control is due to image analysis artefacts while the system equilibrates. A representative experiment of 3 biological repeats is shown. Error bars represent standard deviation of 12 technical replicates.

## Live-imaging rate-of-kill for clinical compounds

A key aim of establishing a live-imaging intracellular *T. cruzi* assay was to enable determination of rate-of-kill (RoK) profiles for anti-trypanosomal compounds with high time-resolution. To achieve this, we imaged infected cells in the presence of test compounds every 3 hours, followed by automated image analysis and plotting of RoK curves using percent infected host cells as the main parameter. Since the imaging instrument only allows acquisition of one field per well in 384-well plates, we averaged data from six replicate wells for each condition to obtain data for a minimum of 2,000 host cells (total number of cells in 6 fields combined). In the first instance, we tested a set of compounds that have been used in the clinical setting

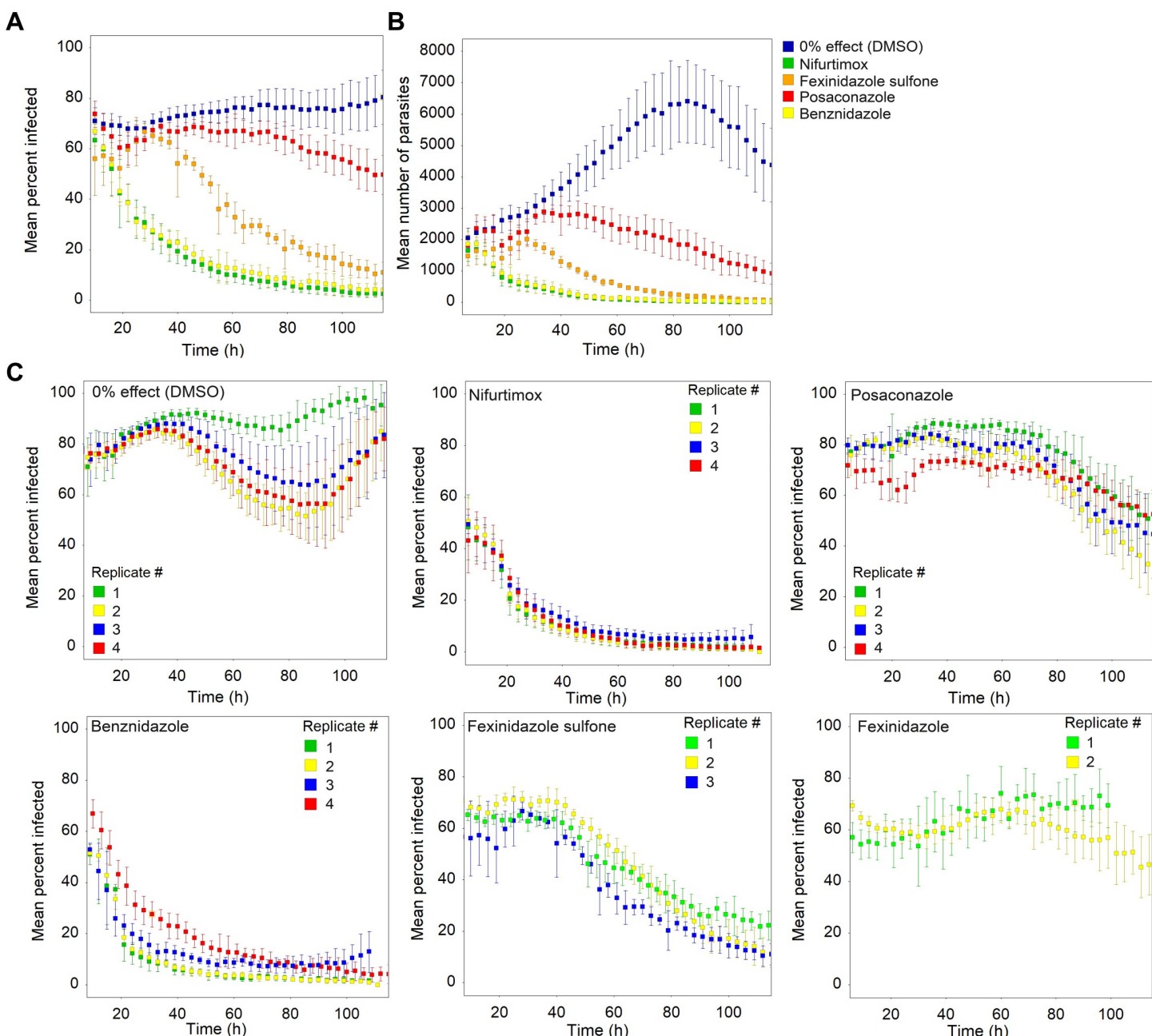

**Fig 2.** Live rate-of-kill curves for reference compounds. A) percent infected host cells versus time, B) mean total number of intracellular amastigotes per well versus time for the same images analysed in A, C) biological replicates of reference compounds plotted as percent infected cells versus time. Concentrations used are as follows: nifurtimox (16 μM), benznidazole (50 μM), posaconazole (0.1 μM), fexinidazole sulfone (50 μM), fexinidazole (50 μM). Error bars in all panels represent standard deviations for 6 technical replicates.

(**Figs 2A** and **S2** and **S2, S3** and **S4 Movies**). For these experiments we tested compounds in dose response mode at eight concentrations to get a detailed picture of the effect of dose on RoK and to identify the concentrations at which maximum RoK was achieved (S2 Fig). Potencies were determined at 120 hours after start of treatment for benznidazole (pEC$_{50}$ 5.8), nifurtimox (pEC$_{50}$ 6.1) and fexinidazole sulfone (pEC$_{50}$ 6) and are in line with published values. Fig 2 shows results for a single concentration corresponding to the maximum RoK observed in the

dose response data. The results show that nifurtimox and benznidazole have an identical RoK profile, that fexinidazole sulfone (the key active metabolite of the drug fexinidazole [21]) has a longer lag phase than benznidazole and nifurtimox and that posaconazole could not clear the parasites from most cells within the time-frame of the assay. Plotting number of amastigotes instead of percent infected cells gives comparable results (**Fig 2B**). As expected, fexinidazole itself showed very little parasite clearance (**Fig 2C**). To assess the reproducibility of the RoK curves, we tested the clinical compounds in multiple independent replicates and found profiles between multiple runs to be very similar (**Fig 2C**). For each independent replicate, both the lag phase and the kill rate were determined (**S1 Table**). When plotted (**S3 Fig**), the high level of reproducibility and how the compounds cluster in short lag/fast RoK and longer lag/slower RoK areas can clearly be seen. Examination of movies from nifurtimox-treated cells revealed that, as expected, the vast majority of host cells are cleared rapidly of parasites. However, some retain a small number of parasites for extended periods of time (>65 hours post compound addition), while other infected host cells die (**S4 Fig and S2 Movie**).

### New chemical entities: Live-imaging RoK and mode-of-action

To explore the diversity of RoK profiles in compounds from our Chagas drug discovery programmes we tested a set of previously published new chemical entities (NCEs) in the live-imaging assay (**Figs S2 and 3A**), including 2,4-diamino-6-methylpyrimidines [14] and 5-amino-1,2,3-triazole-4-carboxamides [12] with unknown mode-of-action (referred to as TCAMS06 and ES08 respectively), divalent transition metal chelating 8-hydroxy-naphthyridines (referred to as B series) [15], 2-amino benzimidazole and aminoquinazolinone inhibitors of *T. cruzi* methionyl tRNA-synthetase (MetRS) [13], chromone inhibitors of *T. cruzi* lysyl tRNA-synthetase (LysRS) [22], and the oxaborole SCYX-6759, which in the related organism *Trypanosoma brucei* acts through inhibition of the nuclear mRNA processing endonuclease, cleavage and polyadenylation specificity factor 3 [16,23,24]. All compounds were tested at four concentrations above their $EC_{50}$ (as generated in our routine screening assay[9]) (**S2 Fig**), and the fastest RoK profile is presented on Fig 3. We observed clear differences in profiles, with differences in lag phase and kill rate. Compounds from the TCAMS06 series, B series and LysRS series as well as the oxaborole SCYX-6759 showed a lag phase of approximately 40 hours post compound exposure, whereas the MetRS and ES08 series compounds only had a minimal lag phase. In terms of maximum rate there were also considerable differences with MetRS01 and ES08 being the fastest acting compounds. For compounds with the same MoA, one would expect similar RoK profiles, provided there are no differences in "cellular pharmacokinetics" (compound influx and efflux, activation, metabolism). To test this hypothesis, multiple compounds from the same chemical series in the RoK assay were profiled (**Fig 3B**). In all cases, compounds from the same series had the same RoK profile.

### Discussion

Detailed understanding of the *in vitro* profile of compounds with anti-*T. cruzi* activity can aid in prioritising these compounds. Rate-of kill data can be used to prioritise faster acting compounds that may offer a higher chance of delivering a short treatment regimen in the clinic. Clearly, prioritisation is multi-factorial and other factors need to be considered as well, including toxicity liabilities and ability to clear persister parasites, which may be independent of the rate-of-kill. In addition, the killing rate as determined in *in vitro* rate-of-kill assays is a frequently-used parameter in dynamic PKPD models that aim to predict *in vivo* and clinical outcomes for infectious diseases such as malaria [25], TB [8] and other bacterial diseases [26–28]. When assessing combination treatment, it may be valuable to determine the rate-of-kill of the

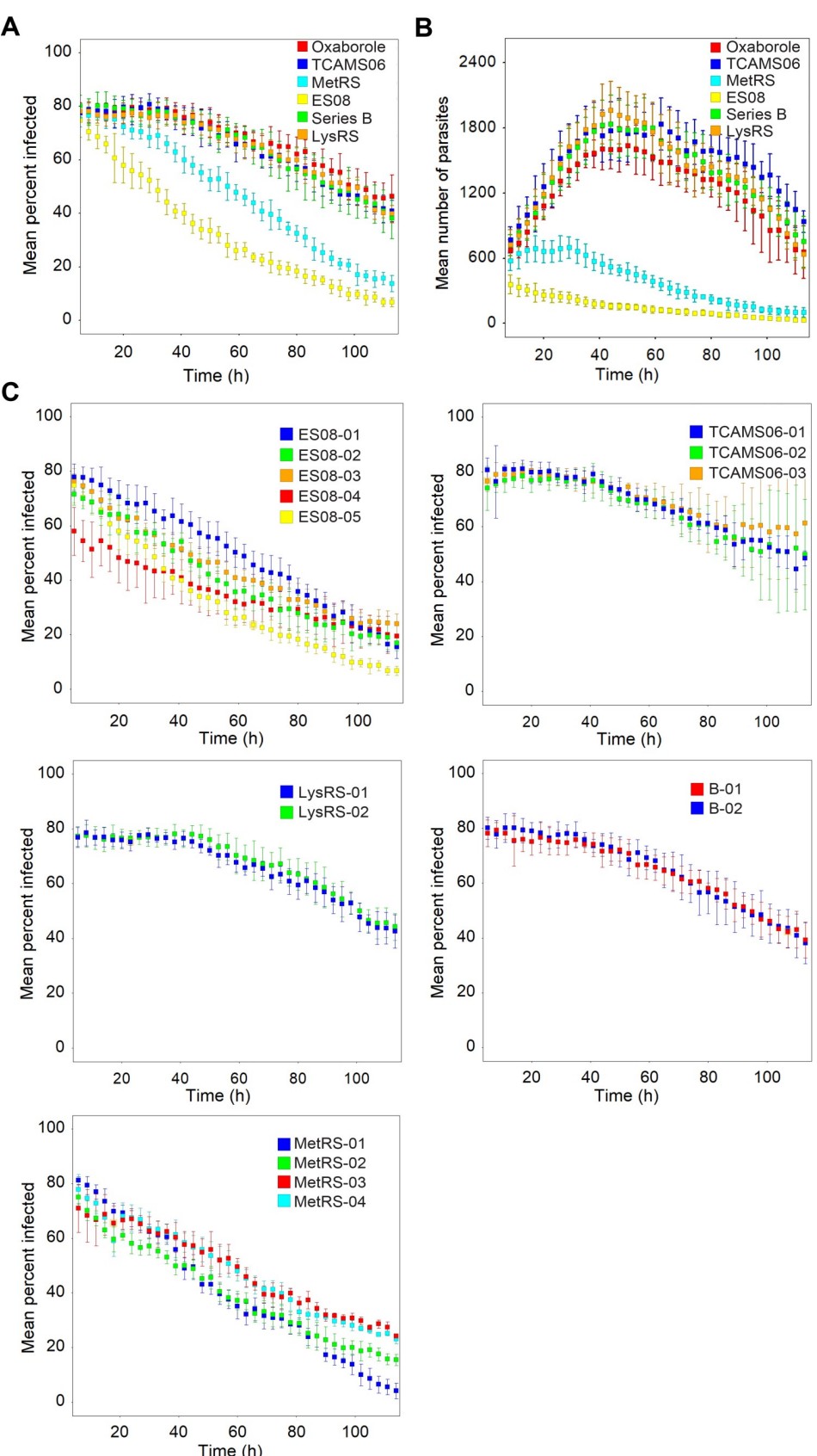

**Fig 3. RoK profiles for NCEs.** A) Percent infected host cells versus time for representative compounds from each hit series, B) mean total number of intracellular amastigotes per well versus time for the same images analysed in A, and C) Percent infected host cells versus time for all tested compounds within each hit series. Profiles are for concentration at which maximum RoK was observed in the four-point dose response data in S2 Fig: oxaborole SCYX-6759 (16.7 μM), TCAMS06 (16.7 μM), MetRS (1.8 μM, higher concentrations showed some host cell toxicity), ES08 (16.7 μM), Series B (16.7 μM) and LysRS (16.7 μM). Error bars represent standard deviation for 6 technical replicates.

combination as well as individual partner drugs, as some combinations may achieve a faster rate-of-kill than any of the partners individually (e.g. [29]). Such combinations would offer the potential of shorter treatment regimens.

Here, we describe a 384-well live-imaging assay as a new tool to provide detailed RoK information to facilitate Chagas disease drug discovery programmes. The key advantages of this assay over existing formats are the superior time resolution (data acquired every three hours or less, compared to every 24 hours for [9,10]) and the ability to monitor the same cells throughout the time-course (compared to different cell populations at each time point in [9,10] or different cell populations every 24 hours [11]). Time resolution for this assay is flexible and can be altered depending on need. Shorter time intervals allow better definition of the RoK profile and more accurate tracking of individual cells throughout the time-course.

We report a fast rate-of-kill for the clinically used Chagas disease drugs nifurtimox and benznidazole, with no observable lag phase. This is in line with previously generated data [9–11]. Also, in accordance with these previous observations, posaconazole is confirmed as a very slow killing compound. Interestingly, a substantial lag phase for fexinidazole sulfone is evident, both when assessing percent infected host cells and total amastigotes, differing from the observations made by Moraes *et al.*[10] where substantial killing was seen after 24h. We confirmed our findings with fexinidazole sulfone in three independent experiments. As expected, very little or no effect on levels of parasite infection was seen in cells treated with fexinidazole, which has been shown to require oxidation to sulfoxide and sulfone metabolites before becoming active [21]. Potential explanations for the difference in RoK profile between our study and that of Moraes and colleagues are the use of different parasite and host cell lines and likely differences in infection burden at start of the assay. The latter factor is particularly relevant as in both assays percent infection of host cells is the primary measurement, thus all parasites need to be cleared from a cell before it is established as uninfected. Therefore, if there is a higher parasite burden in our assay, compared to the published data, then a lag phase is more likely to be observed. The difference in RoK profile that we observe between nifurtimox/benznidazole and fexinidazole sulfone is noteworthy and suggests that while all these compounds are all activated by *T. cruzi* type I nitroreductase [30], their downstream MoA likely differs.

In addition to testing clinical compounds, we profiled a set of compounds from previously published Chagas drug discovery series. As could be expected, compounds from different series showed a range of different profiles, both in terms of lag phase and RoK and this type of data can be used to prioritise fast-killing compounds. To assess the extent that RoK profiles vary within individual compound series, multiple exemplars of each series were profiled and demonstrating that RoK profiles are consistent within a series. This is consistent with the MoA, a key determinant of the RoK profile, being the same for all the compounds within the series. Conversely, RoK profile changes within a compound series could be indicative of a compound that is drifting off-target. Frequent determination of RoK profiles during a phenotypic drug discovery programme can thus help to detect any potential shifts in MoA. However, it should be noted that structurally unrelated compounds demonstrating similar RoK profiles cannot be assumed to act via the same MoA.

The oxaborole SCYX-6759 (AN-4169) showed a relatively slow RoK profile. Moraes et al. previously tested this compound in their RoK assay [10] and found it to be much faster at clearing parasites compared to our results. Likely explanations are as above for fexinidazole sulfone. The discrepancies between these methodologies are important as they show that assay setup can have a major impact on the profiles reported and that it can be difficult to directly compare profiles generated in different assays. It is also important to consider that the RoK profile is not only dependent on the MoA, but also on the intracellular concentration of the compound (dependent on solubility, permeability, active transport etc) and its subcellular localisation.

Close inspection of our imaging data revealed that following nifurtimox treatment some infected host cells die, whereas other host cells were cleared of parasites or retained very small number of parasites. Cell death of infected cells during treatment may indicate that the cellular stress associated with killing intracellular parasites can be detrimental to the host cell. For compounds that demonstrate host cell toxicity in intracellular parasite assays it is thus important to test if the toxicity is also shown against uninfected cells. Our data also indicates that a small number of parasites remain after 65 hours of nifurtimox treatment in line with reports on spontaneous or drug-induced persister forms [19,31,32].

In conclusion, here we present a powerful live-imaging rate-of-kill assay, that not only generates high temporal resolution rate-of-kill data to help prioritise compounds series, but also gives insights that cannot be obtained with previous methods.

## Supporting information

**S1 Text. E2-Crimson gene sequence optimised for expression in *T. cruzi***
(DOCX)

**S1 Fig. Flow cytometry histograms of the *Tc-X10/7-E2Crimson* cell line.** A: comparison of control wild type epimastigotes and *Tc-X10/7-E2Crimson* clone C1 epimastigotes. Over 99% of the *Tc-X10/7-E2Crimson*-expressing epimastigote population is positive for *E2Crimson*. B: comparison of control wild type trypomastigotes and *Tc-X10/7-E2Crimson* clone C1 trypomastigotes. 92% of the *Tc-X10/7-E2Crimson*-expressing trypomastigotes population is positive for *E2Crimson*.
(DOCX)

**S2 Fig.** Live rate-of-kill curves for A) reference compounds and B) representative compounds from each hit series plotted as percent infected host cells versus time. Error bars in all panels represent standard deviations for 6 technical replicates. For compounds were all concentrations showed parasite reduction the DMSO vehicle control plot is also included (blue).
(DOCX)

**S3 Fig. Plot showing lag phase and maximum kill rate for clinical compounds (from data in Table S1)**
(DOCX)

**S4 Fig.** Images from rate-of-kill experiments illustrating cell death of infected host cells (A) and small number of residual parasites after 65 hours of treatment with 16μM Nifurtimox (B). Green fluorescence: H9C2 host cell nuclei, red fluorescence: *T. cruzi* parasites. All images in panel A are for the same field of view. Arrows in panel A point to the same dying cell in each image. Arrows in panel B point to residual parasites.
(DOCX)

**S1 Table. Quantification of RoK profiles: slope and lag phase were determined for each biological replicate, as described in the methods section.**
(DOCX)

**S1 Movie. Fluorescence time-lapse movie generated with the live-imaging rate-of-kill protocol for DMSO treated cells.** Parasites are red, host cell nuclei green.
(WMV)

**S2 Movie. Time-lapse movie generated with the live-imaging rate-of-kill protocol for cells treated with 16μM nifurtimox.** Parasites are red, host cell nuclei green.
(WMV)

**S3 Movie. Time-lapse movie generated with the live-imaging rate-of-kill protocol for cells treated with 50μM benznidazole.** Parasites are red, host cell nuclei green.
(WMV)

**S4 Movie. Time-lapse movie generated with the live-imaging rate-of-kill protocol for cells treated with 3.3nM posaconazole.** Parasites are red, host cell nuclei green.
(WMV)

## Acknowledgments

Thanks to Lorna MacLean for training, advice and helpful discussions and to the *W*CAIR compound handling, data management and lab management teams.

## Author Contributions

**Conceptualization:** Nina Svensen, Manu De Rycker.

**Data curation:** David W. Gray, Manu De Rycker.

**Formal analysis:** Nina Svensen, Susan Wyllie, Manu De Rycker.

**Funding acquisition:** David W. Gray, Manu De Rycker.

**Investigation:** Nina Svensen, Susan Wyllie.

**Methodology:** Nina Svensen, Susan Wyllie.

**Project administration:** David W. Gray, Manu De Rycker.

**Supervision:** David W. Gray, Manu De Rycker.

**Validation:** David W. Gray, Manu De Rycker.

**Visualization:** Nina Svensen.

**Writing – original draft:** Nina Svensen, Manu De Rycker.

**Writing – review & editing:** Nina Svensen, Susan Wyllie, David W. Gray, Manu De Rycker.

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
