## [Decision Letter · Decision Letter 0]

28 Jun 2021

Dear Dr De Rycker,

Thank you very much for submitting your manuscript "Live-imaging rate-of-kill compound profiling for Chagas disease drug discovery with a new automated high-content assay." for consideration at PLOS Neglected Tropical Diseases. As with all papers reviewed by the journal, your manuscript was reviewed by members of the editorial board and by several independent reviewers. In light of the reviews (below this email), we would like to invite the resubmission of a significantly-revised version that takes into account the reviewers' comments. 

We cannot make any decision about publication until we have seen the revised manuscript and your response to the reviewers' comments. Your revised manuscript is also likely to be sent to reviewers for further evaluation.

Sincerely,

Renata Rosito Tonelli, PhD

Associate Editor

Helen Price

Deputy Editor

Reviewer's Responses to Questions

**Key Review Criteria Required for Acceptance?**

**Methods**

-Are the objectives of the study clearly articulated with a clear testable hypothesis stated?

-Is the study design appropriate to address the stated objectives?

-Is the population clearly described and appropriate for the hypothesis being tested?

-Is the sample size sufficient to ensure adequate power to address the hypothesis being tested?

-Were correct statistical analysis used to support conclusions?

-Are there concerns about ethical or regulatory requirements being met?

Reviewer #1: General comments:

- Objectives were clearly stated

- Study design addresses partially the rate of kill data to identify fast killing compounds. It is challenging to compare across compounds given single concentration used in the study. A complete dose response would have given better clarity for comparison across the compounds. 

- Study fails to justify why only fast acting cpds need to be prioritized. 

- claims of rate of kill data provided being helpful for PKPD modeling and combination strategy lacks justification and data to support these claims.

Specific comments:

- A section on image analysis algorithm needs more clarifications. 

o Is image pre-processing performed?

o How was demarcation of host cell cytoplasm done?

o Was there any size and/or intensity threshold used for identifying true parasites versus background (and/or host cells). 

o Does the algorithm only measure intracellular amastigote or trypomastigote as well? 

o How were percentage of infected cells calculated? 

o Approximately, what percentage of a 384 well plate was imaged for every time point?. 

o What was the host cell monolayer confluence, were there specific sections of wells imaged, like the center of the well / edge?

Reviewer #2: yes to all

**Results**

-Does the analysis presented match the analysis plan?

-Are the results clearly and completely presented?

-Are the figures (Tables, Images) of sufficient quality for clarity?

Reviewer #1: General comments

- Single concentration of compound was used to generate the time-course data. The concentrations used were arbitrary. Authors may have to explain the reasoning behind the concentration selected for rate of kill analysis. 

- The EC50 data for this specific parasite in this cell line is needed. EC50 would be great starting point to compare compounds across multiple series.

- One possibility is to generate EC50 or EC90 for all the compounds using 10 point dose response. Use atleast 3 concentration 3X EC50, 10X EC50 and 25X EC50 for rate of kill for all compounds. This might help in interpreting data and also comparison across compounds become more meaningful 

- It would be critical to generate rate of kill using varying concentration of compound. This would give more confidence, if the compounds show time or concentration dependent kill.

- Authors describe that the rate of kill experiments generated would help in PKPD analysis and Combination. But the data generated falls short of providing guidance for PKPD and combination, as the data generated is single point

Specific comments:

- kindly clarify the timelines mentioned, for example, in lines 175 and 179, the authors mention at least four days and day 3 to 4 respectively, which I assume is the time post seeding in 384 well plates (from time 0 in Figure 1A) and not from time of infection. (similar to line 186 which clarifies this timeline clearly)

- Line 182: explain briefly, how the level of infection, i.e., cells infected calculated?

- Lines 206-207: authors mention “we averaged data from six replicate wells for each condition to obtain data for a minimum of 2,000 host cells”. Does this mean each well with at least 2,000 host cells was considered a replicate with 6 such wells / replicates analyzed, or minimum 2000 cells were considered after combining 6 wells? 

- Lines 209-210: How were the saturating concentrations for each compound determined? Is his based on a standard dose response assay – is it a multiple of EC50 or EC90. This is critical since all compounds have been tested only at a single concentration.

- For Figure 3. For new chemical matters it would be nice to show the amastigote over time graph (similar to Fig 2 B)

- Fig 3: ES08 and MetRS have the fastest and second fastest rate of kill but still slower than benznidazole and nifurtimox (as seen in Fig.2). Authors can comment on how to use the data from this assay for the NCE series prioritization?

- Fig. 3. All cpds were tested at 16.7 uM. It will be good to know the relative concentration to their respective EC50. This would help in understanding the Rate of kill pattern relative to the EC50.

- Lines 220-223 (and Fig. S3 and Movie S2): The parasites numbers are cleared to the limit of detection in nifurtimox treatment as show in Fig 2 in the current assay format. A similar Rate of Kill movie for Posaconazole and Benznidazole would be highly beneficial. It will be interesting to see, if Posaconazole arrests replication (static) or even fails to block replication visually.

Reviewer #2: yes to all (except missing caption Table S2; see below)

**Conclusions**

-Are the conclusions supported by the data presented?

-Are the limitations of analysis clearly described?

-Do the authors discuss how these data can be helpful to advance our understanding of the topic under study?

-Is public health relevance addressed?

Reviewer #1: General comments:

Authors have described the live monitoring assay, which specifically address the drawbacks of currently available RoK experiments such as low time resolution and track same cell population over time. But the usefulness of RoK using single concentration of compounds is insufficient to contribute to compound prioritization, comparison across different series of cpds and also within the same series. The behavior of kill pattern might change depending on the concentration of individual compound. The interpretation of same series having same pattern might be too strong for the data presented.

Specific comments

- Lines 294 to 301: Oxaboroles have shown be fast acting against T. cruzi by Moraes et. al. (as discussed by authors) and other trypanosomes. Moraes et al publication used multiple concentrations and showed fast kill. Since the current study is only using single concentration , comparison is challenging andmay be less meaningful. 

- Lines 403 to 406 - Fig. S1: also include information on percentage of host cells and parasites expressing the fluorescent tags

- A live imaging assay was recently published as briefly mentioned by the authors as well (Fesser AF et al. PLoS Negl Trop Dis 14(7): e0008487). The authors assay is significantly improved compared to the previous published data, however, the authors do not discuss their data in comparison this recently published live imaging assay.

- One of the conclusion from the assay is that the compounds from the same series behave similarly in the live imaging assay. Although it is looking similar for NCEs, it may be advisible to test 1-2 more azoles and check, if they behave similar to Posaconazole. (as tested by authors in their previous rate-of-kill assay in MacLean LM et al. PLoS Negl Trop Dis 10(4): e0004584).

Reviewer #2: yes to all

**Editorial and Data Presentation Modifications?**

Reviewer #1: None

Reviewer #2: 1. Data is the plural of datum. Therefore "allow" (Line 26), "these" (Line 187), "show" (Line 210), "indicate" (line 307).

2. Table S1. The caption is missing. Please clarify in the caption whether "Slope" refers to the maximal slope and how the "Lag phase" was calculated.

**Summary and General Comments**

Reviewer #1: The authors have developed a live imaging assay to determine the rate of kill of compounds against the Chagas T. cruzi intracellular parasite stage. For the assay, fluorescent parasites and host cell numbers were tracked and quantified over time with a predominant focus on parasites as the host cells do not replicate in the measured time. Using the assay, clinical compounds were profiled at a single concentration along with a few more previous published new chemical entities for comparison. The assay is robust, reproducible with valuable data on rate of kill of compounds.

But the interpretations are made based on limited single concentration of compound testing could be over-interpreting the data obtained.

Reviewer #2: Svensen et al. describe an elegant new in vitro assay for monitoring the action of test compounds against intracellular T. cruzi in real time and for quantifying the rate-of-kill. There are several improvements as compared to previously published assays (expression of nucleus-restricted GFP in the host cells, reduced level of background fluorescence). The method and the results are described very clearly, and thus this manuscript represents a substantial advance in the field of drug discovery for Chagas disease.

PLOS authors have the option to publish the peer review history of their article (what does this mean?). If published, this will include your full peer review and any attached files.

Reviewer #1: No

Reviewer #2: No
---

## [Decision Letter · Decision Letter 1]

4 Oct 2021

Dear Dr. Manu De Rycker 

We are pleased to inform you that your manuscript 'Live-imaging rate-of-kill compound profiling for Chagas disease drug discovery with a new automated high-content assay.' has been provisionally accepted for publication in PLOS Neglected Tropical Diseases.

Best regards,

Renata Rosito Tonelli, PhD

Associate Editor

Helen Price

Deputy Editor

Reviewer's Responses to Questions

**Key Review Criteria Required for Acceptance?**

**Methods**

-Are the objectives of the study clearly articulated with a clear testable hypothesis stated?

-Is the study design appropriate to address the stated objectives?

-Is the population clearly described and appropriate for the hypothesis being tested?

-Is the sample size sufficient to ensure adequate power to address the hypothesis being tested?

-Were correct statistical analysis used to support conclusions?

-Are there concerns about ethical or regulatory requirements being met?

Reviewer #1: Yes to all

Reviewer #2: -

**Results**

-Does the analysis presented match the analysis plan?

-Are the results clearly and completely presented?

-Are the figures (Tables, Images) of sufficient quality for clarity?

Reviewer #1: Yes to all

Reviewer #2: -

**Conclusions**

-Are the conclusions supported by the data presented?

-Are the limitations of analysis clearly described?

-Do the authors discuss how these data can be helpful to advance our understanding of the topic under study?

-Is public health relevance addressed?

Reviewer #1: Yes to all

Reviewer #2: -

**Editorial and Data Presentation Modifications?**

Reviewer #1: Accept

Reviewer #2: -

**Summary and General Comments**

Reviewer #1: Authors have addressed all the queries adequately.

Reviewer #2: Line 342 - data is still plural: "indicate"

PLOS authors have the option to publish the peer review history of their article (what does this mean?). If published, this will include your full peer review and any attached files.

Reviewer #1: No

Reviewer #2: No

---

## [Editor Report · Acceptance letter]

8 Oct 2021

Dear Dr De Rycker,

We are delighted to inform you that your manuscript, "Live-imaging rate-of-kill compound profiling for Chagas disease drug discovery with a new automated high-content assay.," has been formally accepted for publication in PLOS Neglected Tropical Diseases.

Best regards,

Shaden Kamhawi

co-Editor-in-Chief

Paul Brindley

co-Editor-in-Chief
